# The First Identification of the Uniqueness and Authentication of Maltese Extra Virgin Olive Oil Using 3D-Fluorescence Spectroscopy Coupled with Multi-Way Data Analysis

**DOI:** 10.3390/foods9040498

**Published:** 2020-04-15

**Authors:** Frederick Lia, Jean Paul Formosa, Marion Zammit-Mangion, Claude Farrugia

**Affiliations:** 1Department of Chemistry, University of Malta, 2080 Msida MSD, Malta; jean.p.formosa.12@um.edu.mt (J.P.F.); claude.farrugia@um.edu.mt (C.F.); 2Department of Physiology and Biochemistry, University of Malta, 2080 Msida MSD, Malta; marion.zammit-mangion@um.edu.mt

**Keywords:** extra virgin olive oils, Maltese islands, 3D-fluorescence, PARAFAC, DN-PLSR

## Abstract

The potential application of multivariate three-way data analysis techniques, namely parallel factor analysis (PARAFAC) and discriminant multi-way partial least squares regression (DN-PLSR), on three-dimensional excitation emission matrix (3D-EEM) fluorescent data were used to identify the uniqueness and authenticity of Maltese extra virgin olive oil (EVOO). A non-negativity constrained PARAFAC model revealed that a four-component model provided the most appropriate solution. Examination of the extracted components in mode 2 and 3 showed that these belonged to different fluorophores present in extra virgin olive oil. Application of linear discriminate analysis (LDA) and binary logistic regression analysis on the concentration of the four extracted fluorophores, showed that it is possible to discriminate Maltese EVOOs from non-Maltese EVOOs. The application of DN-PLSR provided superior means for discrimination of Maltese EVOOs. Further inspection of the extracted latent variables and their variable importance plots (VIPs) provided strong proof of the existence of four types of fluorophores present in EVOOs and their potential application for the discrimination of Maltese EVOOs.

## 1. Introduction

The application of a single excitation wavelength for the measurement of multiple emissions is inappropriate for assessing the diversity of the different fluorophores present in an olive oil sample, due to highly overlapping fluorescence bands from multiple fluorophores. This problem can be solved through the application of synchronous spectroscopy or multidimensional measurements, assessing the emission spectra at different excitation wavelengths. Synchronous excitation-emission fluorescence spectroscopy (SEEFS), along with 3D- fluorescence spectroscopy, is nowadays accepted to be more suitable for the analysis of complex multi-component samples than conventional fluorescence spectroscopy. The application of the multicomponent fluorescent systems initially proposed by Lloyd [1] have been successfully employed in the characterization and discrimination of edible oils [2,3,4] and between different categories of olive oil [5]. Moreover, the application of multicomponent fluorescence spectroscopy has also been extended to the detection of adulterants present in virgin olive oil [6,7], in the determination of the extent of oxidation in olive oil [8], and in the determination of phenolic compounds and antioxidants in olive oils under different irrigation treatments [9]. The application of total synchronous fluorescence spectroscopy and excitation-emission fluorescence spectroscopy have also been successfully employed by Sikorska et al. [10] to monitor changes in the olive oil during storage. Apart from extra virgin olive oils (EVOOs), the application of multivariate models in combination with fluorescence spectroscopy has been successfully applied in a number of studies for analysis, characterisation and authentication of food products [11] such as for the classification of wine [12] and Sherry vinegar [13], detection of pesticides and other impurities in honey [14,15,16], screening for antioxidant compounds [17] and assessing thermal degradation in oils [18].

In standard multivariate analysis the data is arranged in a two-way structure, a matrix of observed variables for each sample, such as in the analysis of Fourier-transform infrared spectroscopy (FT-IR), nuclear magnetic resonance (NMR), direct infusion mass spectroscopy (DI-MS) and SEEF data, where the absorbance is determined at set wavelength intervals. This data can be directly analysed through numerous bilinear multivariate techniques such as principle component analysis and different forms of discriminate analysis. Nonetheless, in the case of 3D data analysis, an additional dimension is required. In the case of an excitation-emission matrix (EEM), each sample needs to be described using the fluorescence emission at several wavelengths for several excitation wavelengths on three separate axes in an array. In simpler terms, the intensity can be represented by three indices: sample number, excitation wavelength and emission wavelength. 

In order to analyze three-dimensional data through a bilinear statistical approach, the application of a rather primordial solution of unfolding of the three-dimensional data is necessary. During the unfolding process, each sample (slice) is reorganised and extracted from a three-dimensional matrix in such a way that it is concatenated with the successive sample. The concatenation of data enables each sample to be described using a single row of variables. However, the rudimentary approach for the analysis of trilinear data is susceptible to producing models that are less robust, less predictive and more complex to interpret whilst requiring increased computational power [19,20]. Fortunately, several statistical models exist which enable the direct analysis of multiway data that can be employed in order to analyse 3D data without the need unfold the matrix. 

In recent years the olive cultivation industry in the Maltese islands has re-emerged, potentially allowing the creation of a niche market for high-quality olive oils produced by the Maltese agribusiness sector. At present, most of the cultivated trees used for oil production are imported, since they are associated with better oil yields, placing the relatively unexploited and yet uncatalogued native olive trees at risk. There are three major identified olive cultivars within the Maltese islands which are thought to be native, namely the ‘Bidni’, ‘Bajda’ and ‘Malti’ [21]. Whilst the ‘Bidni’ and ‘Bajda’ are monocultivars, the ‘Malti’ is thought to be made up of several ancient varieties which are geographically isolated from each other [22]. Furthermore, recent studies have shown that Maltese EVOOs have a significantly different phenolic composition and mineral composition [23,24]. In this study, a variety of olive oils selected from different areas around the Maltese islands and countries around the Mediterranean were studied. The application of parallel factor analysis (PARAFAC) and discriminant multi-way partial least squares regression (DN-PLS) were employed in order to differentiate Maltese EVOOs from EVOOs derived from other countries within the Mediterranean region using full EEM, thus developing a quick, easy and cost-saving verification method for the origin of EVOOs from the Maltese islands, paving the path for the application of protected designation of origin.

## 2. Materials and Methods

### 2.1. Sample Preparation

For this preliminary study, a total of 65 extra virgin olive oil samples were collected from the Maltese islands over four harvest seasons from 2013–2016 and from other neighboring Mediterranean countries. The samples were all taken from different oil producers so as to cover a representative sample of the Maltese islands in terms of pedological and microclimatic conditions, whilst also accounting for manufacturing techniques and the different presses employed. Foreign olive oils obtained were bought with a protected designation of origin in order to ensure traceability of the product. All the samples were stored at 4 °C in the absence of light prior to analysis.

### 2.2. EEM Spectra Acquisition 

A three-dimensional (3D) matrix excitation-emission matrix (EEM) was obtained for each sample using a Jasco FP-8300 fluorescence spectrophotometer. Both the excitation and the emission bandwidths were set at 5 nm for a measurement range between 210 to 750 nm. The acquisition interval and the integration time were maintained at 0.5 nm and 10 ms, respectively, with a scan speed of 5000 nm·min^−1^. The oil samples were examined by means of right-angle geometry. 

### 2.3. The PARAFAC Model

The easiest way to describe PARAFAC is that it is an extension of a bilinear PCA. PARAFAC decomposes the cubed data into three loading matrices, A(I,F), B(J,F) and C(K,F), each corresponding to the modes/directions of the data cube with elements a_if_, b_jf_ and c_kf_, respectively. The PARAFAC model tries to minimize the sum of squares of the residuals denoted as e_ijk_ where F denotes the number of factors.
(1)$$X_{\mathrm{ijk}}=\overset{F}{\underset{f=1}{\sum }}a_{\mathrm{if}}b_{\mathrm{if}}c_{\mathrm{kf}}+e_{\mathrm{ijk}}$$

The decomposition is made into trilinear components. In comparison with other bilinear models where each factor consists of one score vector and one loading vector, in PARAFAC each factor consists of three loading vectors (a,b,c). For PARAFAC analysis the data from the excitation range of 220–240 nm were removed from the 3D-EEM due to anomalous noise and instrumental artefacts present. Furthermore, areas with emission wavelengths smaller than or equal to excitation wavelengths were set to a value of zero as expected by the laws of physics, namely that a fluorophore cannot emit light of higher energy than the source of excitation. The values along the diagonal of the EEM where λ_ex_ = λ_em_, and the values right next to this diagonal (λ_ex_ < λ_em_) were set to 0’. A blank spectrum (iso-octane) was recorded with three accumulations and subtracted from all sample spectra prior to multi-way modelling. The optimal number of components was determined using split-half validation analysis combined with the core consistency, together with the % of the explained variance and the residuals. Spilt-half validation analysis [25] takes advantage of the uniqueness property of the PARAFAC model, stating that the same B and C loadings should be found in different subsets of the data. During split-half validation analysis, the data is split into two independent sets and a PARAFAC model is built independently. The second most important diagnostic for the determination of the optimal number of components during PARAFAC analysis is the core consistency. This parameter determines the appropriateness of the PARAFAC solution. One major limitation of the core consistency is that unless coupled with other parameters, the core consistency will not definitively show which number of components is ideal, as it only provides information about whether the model is valid or not (overfitted). Well-described models will have a high core consistency with a value close to 100, while significantly lower values serve as an indication of the presence of redundant components [26,27]. The cut-off point to identify the optimal number of components and hence the goodness of the model is to find a balance between the core consistency and explained and residual variation, together with the split-half validation. After extracting the optimal number of components, LDA was carried out on the extracted scores from mode 1 (sample mode) for the PARAFAC model with the optimum number of components being carried out using The Unscrambler X 10.3 (CAMO Software, Camo Analytics, Oslo, Norway). Univariate analysis and logistic regression analysis were carried out using SPSS version 22 (IBM Corp, SPSS Inc., Chicago, IL, USA).

PARAFAC and DN-PLS modelling was performed using the ‘N-way’ MATLAB toolbox from Eigenvector. No data pre-processing treatments were applied to the input data array, and non-negative constraints were applied to the PARAFAC model in all modes as negative spectra or concentrations were not expected. The convergence criteria were set to a minimum tolerance of 1 × 10^−10^ and a maximum analysis time of 1 h using singular value decomposition SVD for model initialization. For each PARAFAC model, it was determined that the convergence criteria with respect to tolerance criteria were met. The optimum number of components was determined by building 10 PARAFAC models, each having a different number of components (1–10), and the optimum model was determined using split-half analysis. Each PARAFAC model was replicated ten-fold, in order to ascertain true convergence. For the model to provide a meaningful solution, constraints were applied during PARAFAC modelling, which included non-negativity and unimodality constraints.

### 2.4. Discriminant Multi-Way Partial Least Squares Regression (DN-PLSR)

Multi-linear or multi-way partial least squares (DN-PLSR) regression can be defined as an extension of the bilinear PLS to multi-way data. In the case of 3D data, the method is assigned as tri-PLS. In this form of DN-PLSR, the 3D matrix, X, is decomposed into a set of triads. A triad is defined as the trilinear equivalent of a bilinear factor which consists one score vector, t, and two weight vectors—one in the second order, called w^J^, and one in the third order called w^K^—in a similar fashion to PARAFAC modelling. The aim of DN-PLSR, however, is to maximize the covariance between the data and the response as in ordinary PLS. The overall model, X, can be expressed by the equation below where the term e_ijk_ represents the error which is not explained by the overall model. Using DN-PLSR as opposed to bilinear PLS on the unfolded multi-way data has the advantages of developing more robust, less complex and more easily interpretable models [27].
(2)$$X_{\mathrm{ijk}}=t_{i}w_{j}^{J}w_{k}^{K}+e_{\mathrm{ijk}}$$

In this model, the response Y matrix, which corresponded to the geographical origins of the EVOOs, is made up of a set of columns, whereby each column represents a class and contains samples that belong to this class, represented as (1), whilst for those that do not were represented as (0). In the special case when there are only two classes, y is a column vector with ones for the samples that belong to one of the classes (EVOOs of non-Maltese origin) and zeros for the samples that belong to the other (EVOOs of Maltese origin). For DN-PLS analysis the data were prepared in a similar fashion as for PARAFAC; however, only the regions between 270 and 510 nm (excitation) and 290 and 575 nm (emission) were used. The optimum number of components was determined by building 15 DN-PLS models, each having a different number of components (1–15), and the optimum model was determined using the prediction accuracy and RMSE error of both the calibration and validation models. Validation of the model was carried out using Venetian blinds cross-validation, which selects every sth sample from the data by making s data splits such that all samples are left out exactly once (*s* = 3). Mean centering was applied in the first mode of the multi-way array, which corresponds to the sample mode prior to DN-PLSR analysis, and all missing values were set to have a 0 value. 

In order to assess the predictability of the PARAFAC and DN-PLSR analysis the scores obtained for mode 1 were subjected to an independent LDA. A Venetian blinds cross-validation was carried out such that the whole dataset was split into two sets: the training and test sets (the former to build the model, the latter to validate its predictability). The Maltese and the non-Maltese samples were grouped in an ascending way so that the first 30 samples would represent Maltese EVOOs, whilst the rest corresponded to non-Maltese EVOOs. A Venetian blinds cross-validation was carried out, which selects every sth sample from the data by making data splits such that all samples are left out exactly once (*s* = 5). This sampling method excluded 20% of the observation so that they would be retained as the testing set. The remaining 80% of the observation was used to build the training set.

## 3. Results

### 3.1. Extraction of PARAFAC Components 

From the results (Table 1) obtained it was determined that a four-component PARAFAC model was the optimum model based on explained variance, residual variance, core consistency and split similarity.

These four components were identified through their characteristic λ_ex_ and λ_em_ maxima, as found in the literature [28,29,30,31,32], as the four major fluorescent compounds found in EVOOs which correspond to chlorophyll compounds, tocopherols, phenolic and oxidised compounds. Figure 1 shows the typical EEM for each PARAFAC component; the colour indicates the typical intensity observed for each of the compounds.

If a four-component model is appropriate in describing the data set, it is reasonable to assume that these four components have maxima appertaining to particular chromophores found in EVOO. The first component was attributed to chlorophylls having an emission band with a maximum at 675 nm (λ_em_), which is associated with the presence of chlorophyll pigments in the samples [31]. The emission profile of the second factor showed a band with a maximum at 525 nm, with an excitation at 325 and 340 nm, and these were assigned to oxidation compounds [31]. This band (λ_em_ = 450–650 nm) slightly overlaps with the 3rd component; however, it is completely absent in the emission profile of the 4th component and 1st component. These results are consistent with the PARAFAC results obtained by Tena et al. [32] and earlier by Guimet et al. [28,29]. In the results obtained by Tena et al. [32] the remaining 3rd component showed a characteristic band with a maximum at 350 nm (λ_em_) and 285 nm (λ_ex_), which was collectively associated with the presence of tocopherols and phenols, as previously identified by Sikorska et al. [10] and Zandomeneghi et al. [33]. However, in this experiment, it was shown that it is possible to distinguish between the 3rd and 4th component.

The 3rd component was identified as belonging to the tocopherols and tocotrienols. The wide emission band was attributed to the presence of the different isomeric forms of the different classes, namely α-β-δ-γ, which was previously identified by Eitenmiller et al. [34]. These compounds have an excitation in the range of 290–297 nm and emission in the range of 386–468 nm. In the case of the 4th component, this was attributed to the presence of phenolic compounds in EVOOs. Tena et al. [32] showed that phenolic compounds belonging to the secoiridoid class (oleuropein) had an excitation at 270 nm and emission at 310 nm, whilst simple phenolic acids (gallic, vanillic, caffeic) and simple phenolic alcohols (tyrosols) shared the same excitation maxima, however the emission spanned from 349 to 457 nm. A similar conclusion was drawn by Cheikhousman et al. [35], whereby it was shown that the excitation and emission maxima obtained at λ_em_ = 380 nm and λ_ex_ = 295 nm agree very well with the respective spectra of α-tocopherols, whilst the hypsochromic shift compared to α-tocopherol observed at λ_em_ = 300 nm and λ_ex_ = 280 nm was attributed to the phenolic compounds.

### 3.2. Linear Discrminate Analysis on Mode 1 of the PARAFAC Compontents PARAFAC-LDA

After the extraction of the four-component model, the concentrations of each identified fluorophore can be determined through the examination of loadings in mode 1. The application of Fisher linear discriminate analysis on Mode 1 (relative concentration shown in Figure 2) of the extracted components showed that 73.0% of the original data and 80.28% of the cross-validated data set were correctly classified. It was shown that after the application of an LDA model to the components extracted using PARAFAC a slight overlap between the two classes was observed. This suggests that PARAFAC can extract components which reflect the class of fluorescent compounds found in EVOO. The concentration of each class of compounds can be used to discriminate between EVOOs of different origin; however, better classification models can be obtained using discriminant multi-way partial least squares regression performance.

### 3.3. Discriminant Multi-Way Partial Least Squares Regression Performance

In comparison to the PARAFAC models obtained, DN-PLSR analysis was shown to be more effective in classifying Maltese and foreign samples, as shown in Table 2. The 12-component model was chosen to be the most suitable model as it had the highest classification rate in the validation stage and a relatively low root mean squared error cross validation (RMSECV) and root mean squared error calibration (RMSEC).

## 4. Discussion

### 4.1. Extraction of the Optimal Number of PARAFAC Components

The extraction of the optimum number of components (factors, i.e., the chemical rank of the data) in PARAFAC is a crucial step. Extracting too few components will result in underfitting; this can be easily spotted via the inspection of the explained variance and the residuals, as an underfitted model tends to have a low explained variance and a high residual variance. On the contrary, the use of too many components will result in overfitting of the model. In this case, the model obtained will have almost a 100% explained variance and very low residual variation. In such cases, the model is not necessarily modelling noise but also modelling factors which are correlated with each other [19]. A number of different methodologies are employed to determine the optimum number of factors of a PARAFAC model. The most common method includes split-half validation analysis combined with the core consistency [30], together with the percentage of the explained variance and the residuals. In the case of residual analysis, the residual variation obtained for the inclusion of each component is analysed, similar to bilinear models. A small drop in the residual variation from one component to another suggests that the component is not explaining much of the variance within the data set and is thus redundant.

From the results obtained it was determined that a four-component PARAFAC model was the optimum model based on explained variance, residual variance, core consistency and split similarity. Similar results were obtained by Guimet et al. [28], whereby a four-component model (explained variance 98.7%) was found to be optimal in distinguishing between commercial samples of virgin and pure olive oils. Furthermore, studies conducted by the same authors in 2005 showed that the application of PARAFAC as a complementary technique for olive oil characterization indicated that the optimal number of factors was three (98.65% explained variance). The difference in the optimal number of PARAFAC components obtained by the same authors can be explained in terms of the initial data inputted. Whilst the first study focused on the discrimination of pure olive oils, the second study carried out in 2005 [29] focused mainly on relating the quality parameters of different EVOO grades to the EEM.

No improvement was observed for the five-component model, even after several replicates. The low value for split similarity for the five-component model was used to discount the model as inappropriate, and thus, the four-component model was chosen as the optimum due to the relatively high core consistency and split similarity (Appendix A).

### 4.2. Application of Univariate and Multivariate Analysis on the PARAFAC Components in Mode 1

Whilst the loading matrices of mode 2 and mode 3 provide information about the excitation and emission spectra of the different fluorophores involved, mode 1 provides information about the concentration of mode 2 and 3 in the samples. In fact, PARAFAC components can be a direct representation of the concentration of chemical constituents within several samples, as demonstrated by Bro [19]. Figure 3 also shows the relative concentrations obtained for a four-component PARAFAC model and how these vary for EVOOs of different geographical origin. Univariate normality testing for the different components revealed that the concentration of the first two components, chlorophyll and oxidised by-products, in EVOOs were consistent with a normal distribution under Shapiro-Wilk’s Normality Test with a *p*-value of 0.2673 and 0.5886 respectively. On the other hand, the 3rd and the 4th component, which correspond to the concentration of tocopherols and phenolic compounds respectively, were found to be inconsistent with a non-parametric distribution, with *p*-values of 0.0039 and <0.0001. Analysis of variance between the Maltese and foreign EVOOs for the first two components was carried out using ANOVA, whilst for the 3rd and 4th component, this was carried out using a non-parametric Kruskal-Wallis test. From the results obtained, it was found that EVOOs of Maltese origin tend to have a marginally significantly higher chlorophyll concentration (*p*-value 0.090* significant at the 90% confidence level), whilst no significant difference (*p*-value 0.419) was observed in the concentration of oxidized by-products between the Maltese and foreign EVOOs. These results suggest that all the samples obtained were fresh and that there were no gross outliers in terms of EVOO oxidation. The non-parametric Kruskal-Wallis test revealed that the Maltese EVOOs had a significantly lower concentration of both the phenolic compounds and tocopherols (*p*-value < 0.001 for both components). These observations suggest that apart from the genetic inference affecting several fluorescent compounds like tocopherols and polyphenols, the pedoclimatic conditions might also be affecting the amount of these constituents in EVOOs.

A linear discriminate analysis (LDA) was carried out on the extracted mode concentrations obtained. Discriminant analysis enables the construction of a predictive model for group membership. The prediction model obtained is constructed using linear combinations of predictor variables. Furthermore, this model also provides insights on those predictor variables which provide the best discrimination between groups. Application of multivariate normality testing on the extracted compounds revealed significant deviation from normality under several multivariate normality tests (Doornik and Hansen omnibus test, Mardia’s, Henze-Zirkler, and Royston *p*-value < 0.0001), and thus, a Fisher LDA method was employed. Through analysis of the discriminate model performance, it was shown that the model obtained was able to correctly classify 73.0% of the original data and 80.28% of the cross-validated data. Analysis of the standardised scoring coefficients and the Pearson’s correlations of each variable with the discriminant function further confirmed that the phenolic and tocopherol compounds are mainly responsible for the discrimination of Maltese EVOOs from foreign EVOOs (Appendix A).

### 4.3. Discrimination of Maltese EVOOs through DN-PLSR

The advantages of using DN-PLSR are mainly its robustness and the high correct classification rates [19,20], even when no data pre-treatment or variable selection is used. Whilst data pre-treatment might improve the prediction rate by removing unnecessary information and instrumental artefacts, transformations of multi-way arrays are very complex, and most transformations which exist for bilinear data or their multi-way equivalents are not readily available for multi-way data. Table 2 illustrates different measures of model fit on using a different number of latent variables, with each parameter represented as an average value and ± 1SD obtained from using 3 different splits. The calibration accuracy and root mean square error of calibration (RMSEC) indicate model performance during the calibration (training) stage, while validation accuracy and root mean square error of cross validation (RMSECV) deal with the performance of the model with regards to the validation samples. These values, along with % explained variance, are indicators of the suitability of the model for prediction.

The loading components are represented in Figure 4, although these did not have distinctive shapes of fluorescence spectra as previously found during the PARAFAC analysis; thus, the likelihood of a distinct chromophore or a set of chromophores which are directly responsible for classification cannot be directly drawn. However, on further inspection it can be observed that in the case of excitation, the loading tends to be higher, in the region of 260–320 nm and 370–450 nm, while in the case of emission these tend to be centered around 300–470 nm and 620–680 nm. Albeit not distinctively, these loadings are in fact reflecting the four different fluorophores present in EVOO previously identified using PARAFAC. Whilst in the case of PARAFAC each fluorophore was described using a single component, in the case of DN-PLSR each fluorophore is described by two or more components, which when added together reveal the fluorescent profile of the fluorophore. It is the complex interaction of these fluorophores together that is being used for the correct classification of the samples.

This was further confirmed on inspection of the VIPs obtained, as shown in Figure 5. A VIP score is a measure of a variable’s importance in the partial least square (PLS) model. It represents the contribution of a variable to the PLS model and is determined by a weighted sum of the squared correlations between the model components and the original variable. A value of less than 0.8 is typically considered to be a small VIP and thus a candidate for deletion from the model [36,37]. Analysis of the VIPs revealed that, in fact, the most discriminating predictors are those associated with the presence of the four distinct fluorophores previously determined during the PARAFAC.

Figure 6 shows the results obtained using independent linear discriminate analysis on the scores obtained in sample mode. The results obtained showed that during the training the LDA models obtained on scores from the three different splits had a higher discriminatory power than those obtained from mode 1 of PARAFAC. In fact, the LDA models had 98–100% correct classification in the training set and a predictability of 92.5–98.0%. This shows that whilst PARAFAC is ideal for the determination of the individual classes of fluorescent compounds found in EVOOs, the application of a four-component model is less discriminate when compared to the discriminate models obtained using scores from DN-PLSR. From the analysis of the loading it was shown that the previously identified classes in the PARAFAC are explained by more than one component extracted from the DN-PLSR, possibly due to variation in the individual compounds making up each class.

## 5. Conclusions

The application of 3D-fluorescence spectroscopy in conjunction with three-way methods has proven to be a useful tool for analysing and interpreting this kind of complex data. Identification of the four detected fluorescent components was fully achieved offering a cheap, fast and reliable way for the discrimination of Maltese EVOOs from non-Maltese EVOOs. Although differing in their discriminatory power between different EVOOs, the underlying concept of four-component fluorophore-based discrimination tends to be corroborated using PARAFAC and DN-PLSR. The application of LDA on the four-component PARAFAC model in mode 1 was able to correctly classify 73.0% of the original data and 80.28% of the cross-validated group, whilst the application of 12 component model in the DN-PLSR was able to correctly classify 98.58% of the original data and 93.18% the cross-validated group. These results suggest that the latter method offered discriminatory potential for the determination of the authenticity of Maltese EVOOs.

## Figures and Tables

**Figure 1 foods-09-00498-f001:**
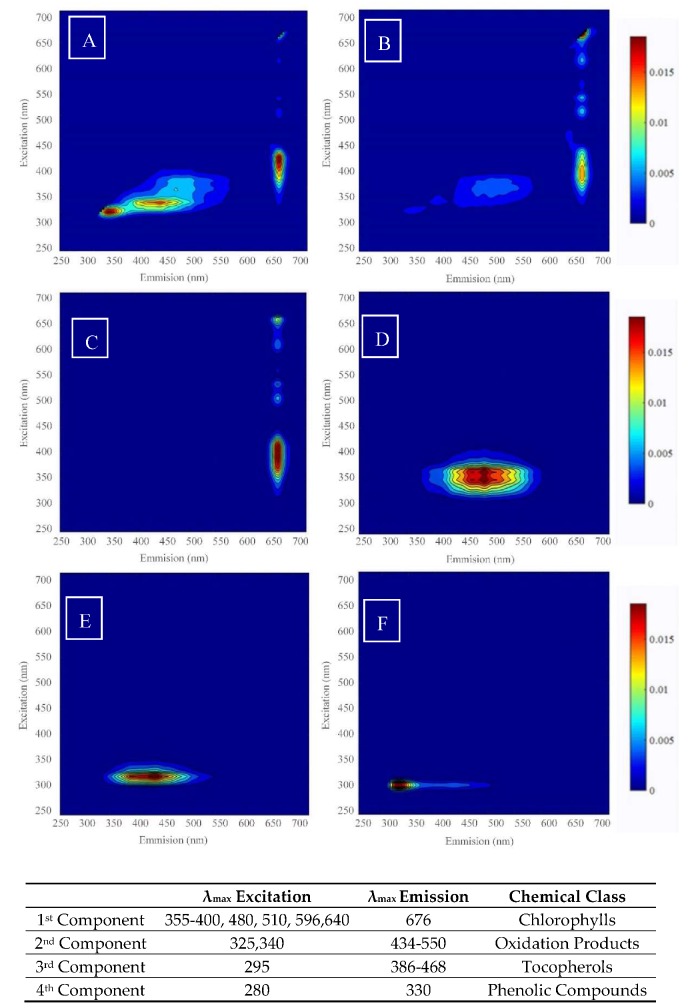
Mode 2 (excitation) and mode 3 (emission) PARAFAC components from four component modes represented as an EEM spectrum. (**A**) EEM of typical Maltese EVOO, (**B**) EEM of typical non-Maltese EVOO, (**C**) Chlorophyll, (**D**) oxidation products, (**E**) tocopherols, (**F**) phenolics.

**Figure 2 foods-09-00498-f002:**
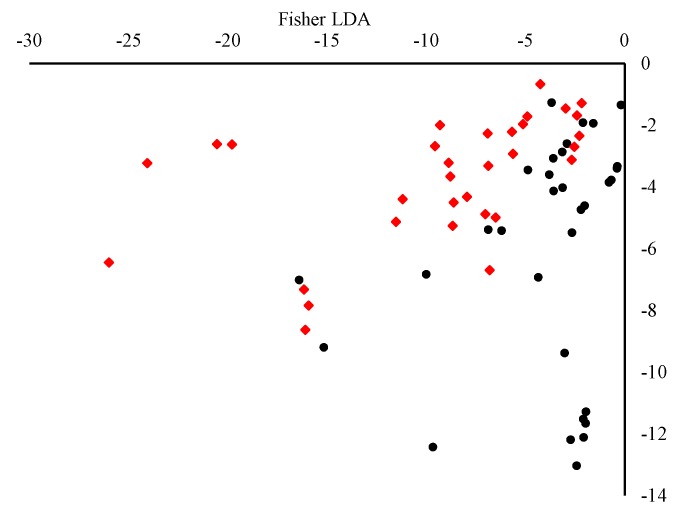
Linear discriminant analysis biplot obtained using non-parametric Fisher type on mode 1 scores of the four-component PARAFAC model. (Black circles) EVOOs of Maltese origin and (red diamonds) EVOOs of non-Maltese origin

**Figure 3 foods-09-00498-f003:**
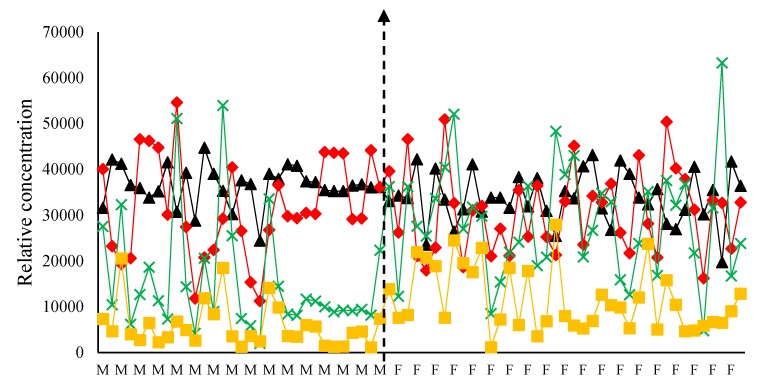
Mode 1 loadings (relative concentrations y-axis) from non-negative constrained four-component PARAFAC model, and how they vary for the different samples (x-axis). Samples of Maltese origin are denoted by the letter M, whilst foreign samples are denoted by the letter F. Black (▲) line indicates the relative concentration of chlorophyll pigments; green line (x) represents the tocopherol content; yellow line (■) represents the phenolic content; red line (♦) represents the oxidation products.

**Figure 4 foods-09-00498-f004:**
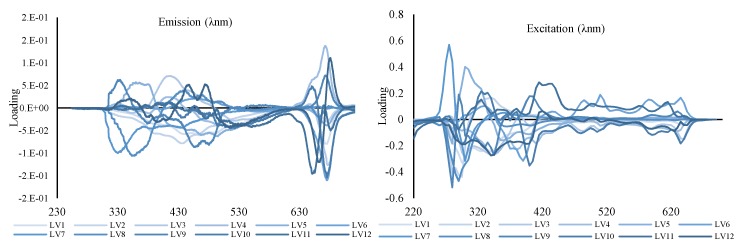
Loading obtained for emission (left) and for excitation (right) using 12 components obtained using DN-PLSR.

**Figure 5 foods-09-00498-f005:**
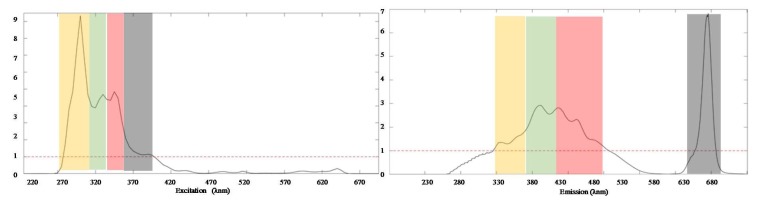
Variable importance plot (VIP) loading of DN-PLSR highlighting the four main regions of importance, as previously identified through PARAFAC, which correspond to the phenolic compounds (yellow), tocopherols (green), oxidised products (red) and chlorophylls (black).

**Figure 6 foods-09-00498-f006:**
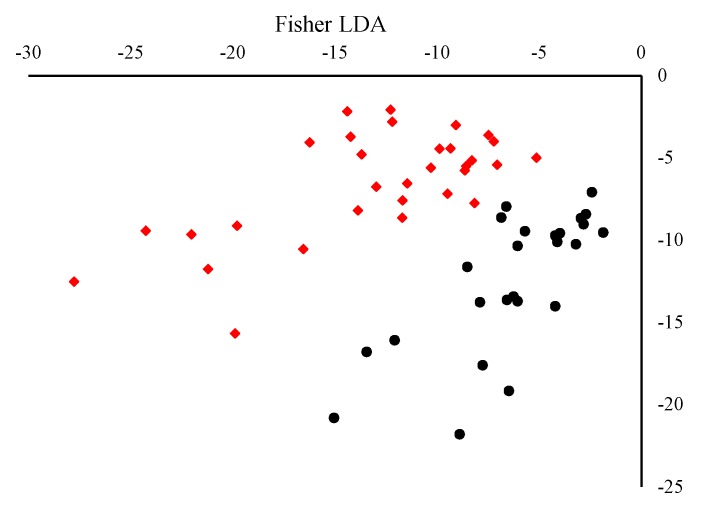
Linear discriminant analysis biplot obtained using non-parametric Fisher type on mode 1 scores of the 12-component DN-PLSR model. (Black circles) EVOOs of Maltese origin and (red diamonds) EVOOs of non-Maltese origin.

**Table 1 foods-09-00498-t001:** Results from non-negative constrained parallel factor analysis (PARAFAC) model. The scaled residual variance, explained variance, core consistency and split similarities are shown for models having between 1 and 9 components. For presentation reasons, the residual variance is scaled to the maximum residual variance.

	Core Consistency %	Explained Variation %	Residual Variation	Split-Half Similarity %
PC1	100.00	62.86	0.952	98.81
PC2	98.80	86.42	0.868	97.60
PC3	90.12	93.12	0.74	96.20
^1^PC4	91.97	95.21	0.627	88.54
PC5	83.93	96.47	0.493	59.18
PC6	−169.51	97.19	0.364	44.40
PC7	−72.12	97.58	0.261	0.00
PC8	−699.24	97.83	0.174	0.00
PC9	−11,859.88	98.21	0	0.00

^1^ The four-component model was deemed to have the optimum number of components.

**Table 2 foods-09-00498-t002:** Compiled output obtained from the analysis of the three different splits carried out in Discriminant Multi-Way Partial Least Squares Regression (DN-PLSR).

No.LV	%Variation X	%Variation Y	Overall %Variation	Training	Validation	RMSEC	RMSECV
1	62.16 ± 0.57	57.97 ± 2.16	69.05 ± 9.18	70.21 ± 8.74	67.14 ± 10.98	0.48 ± 0.02	0.45 ± 0.06
2	73.86 ± 1.13	68.62 ± 2.07	77.14 ± 3.78	78.72 ± 6.49	74.95 ± 7.34	0.43 ± 0.02	0.44 ± 0.04
3	77.93 ± 1.97	70.60 ± 1.96	78.10 ± 3.60	78.01 ± 6.38	76.21 ± 6.17	0.41 ± 0.02	0.39 ± 0.06
4	85.18 ± 2.90	78.37 ± 1.74	80.95 ± 5.02	83.69 ± 5.38	78.87 ± 8.78	0.37 ± 0.02	0.40 ± 0.07
5	90.91 ± 1.48	81.83 ± 4.44	83.81 ± 4.36	89.36 ± 4.43	81.62 ± 10.82	0.33 ± 0.04	0.41 ± 0.08
6	93.25 ± 1.13	85.38 ± 3.55	87.62 ± 4.59	92.91 ± 4.58	85.45 ± 9.33	0.30 ± 0.03	0.40 ± 0.09
7	94.66 ± 1.24	87.29 ± 2.78	87.62 ± 5.95	93.62 ± 4.84	85.48 ± 12.20	0.28 ± 0.03	0.39 ± 0.08
8	95.13 ± 1.20	88.66 ± 2.14	88.57 ± 4.29	93.62 ± 3.94	86.78 ± 6.85	0.26 ± 0.02	0.39 ± 0.06
9	95.40 ± 1.26	89.67 ± 1.87	90.00 ± 3.78	94.33 ± 3.41	88.37 ± 6.85	0.25 ± 0.02	0.40 ± 0.02
10	95.67 ± 1.23	90.90 ± 1.93	91.90 ± 5.77	96.45 ± 4.29	90.06 ± 5.46	0.23 ± 0.02	0.40 ± 0.02
11	95.90 ± 1.27	91.98 ± 1.17	93.81 ± 4.36	98.58 ± 3.78	91.79 ± 5.49	0.22 ± 0.02	0.40 ± 0.02
^1^12	96.62 ± 1.14	93.00 ± 1.15	94.76 ± 3.30	98.58 ± 2.94	93.18 ± 5.19	0.20 ± 0.02	0.39 ± 0.01
13	97.12 ± 0.64	94.06 ± 0.92	93.81 ± 2.97	98.58 ± 3.24	92.25 ± 5.19	0.19 ± 0.02	0.40 ± 0.00
14	97.52 ± 0.76	94.88 ± 0.78	92.86 ± 4.29	98.58 ± 3.71	91.30 ± 5.42	0.17 ± 0.01	0.43 ± 0.03
15	97.71 ± 0.64	95.77 ± 0.45	93.33 ± 4.36	99.29 ± 3.87	91.76 ± 5.66	0.16 ± 0.01	0.42 ± 0.05
16	97.86 ± 0.60	96.32 ± 0.46	92.86 ± 2.86	100.00 ± 4.76	91.11 ± 7.23	0.15 ± 0.01	0.42 ± 0.06
17	98.03 ± 0.62	96.64 ± 0.67	93.81 ± 2.18	100.00 ± 5.80	92.34 ± 6.14	0.13 ± 0.01	0.42 ± 0.07
18	98.17 ± 0.68	96.89 ± 0.71	94.76 ± 1.65	100.00 ± 4.39	93.51 ± 5.16	0.13 ± 0.02	0.41 ± 0.06
19	98.40 ± 0.45	97.47 ± 0.61	93.33 ± 3.60	100.00 ± 3.99	92.00 ± 7.09	0.11 ± 0.02	0.42 ± 0.03
20	98.56 ± 0.37	97.91 ± 0.46	61.43 ± 27.22	63.83 ± 27.29	66.89 ± 25.87	0.10 ± 0.01	0.62 ± 0.19

^1^ The 12 latent variable model was deemed to have the optimum DN-PLSR model performance.

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
