# Peer review of "The First Identification of the Uniqueness and Authentication of Maltese Extra Virgin Olive Oil Using 3D-Fluorescence Spectroscopy Coupled with Multi-Way Data Analysis"

_foods, 2020, doi:10.3390/foods9040498_

Round 1

Reviewer 1 Report

Comments on “The first identification of the uniqueness and authentication of Maltese Extra virgin olive oil using 3D-fluorescence spectroscopy coupled with multi-way data analysis”

The authors open the Introduction with limitations of certain spectroscopic approaches. The paper needs more context, motivation to open the introduction. Why is this study needed, what is the state of the art prior to this study?

There are no recent literature references in this paper, the latest being 2008. Has anyone used these methods more recently? Can you point to similar, more recent studies? Similar either in terms of the statistical methods used (in food provenance/authenticity) or in studies of food oils.

Page 2, Line 59, “…thar are less robust…” change to “...that are less robust…”

Figure 1. I cannot read the wavelength scales. Use larger font. Render in higher resolution pdf. The image quality in the pdf manuscript I downloaded is terrible, not fit for publication. Maybe my download corrupted the graphical quality, but I don't think so.

Very little discussion of Figure 2. Is it really needed for the paper? Besides it seems to show that the two sample types are not well discriminated (only a handful of data values in each set are clearly outside the bounds of the other set).

Line 90: Equation should not have X_ijk at the end next to the error term e_ijk (see PARAFAC wiki page, for example)

Line 98: “The values along the diagonal…” Diagonal of what matrix or array? Be specific.

Line 100-101: Either explain terms “split-half validation analysis” and “core consistency” here or note that they are explained later.

Line 102: Balance between which two parameters? Also I think you mean something like “measures” rather than “parameters.” Split-half validation, core consistency, etc. are measures of “goodness of a model,” not parameters of a model.

Line 105: What is meant by “mode” here? Could just note that modes are explained later.

Line 131-2: Where did the variable Y come from? Do you mean X? And are you saying that X is an array of 0s and 1s? This description of response matrix and classes is confusing. I doubt the comment about the special case of only two classes is needed; clearly that doesn’t come up in your analysis.

Line 142: What is meant by “sample mode”? That terms does not seem to have been defined.

Table 1, column 4: Residual variation increasing with increasing number of components seems odd to me; perhaps I misunderstand what you mean by “residual variation.” I would expect the variance in residuals to decrease as the model fit improves.

Line 154-155: It may be worth noting that how these four components were identified as specific fluorescent compounds is described later in the manuscript.

Figure 1: This figure is not called out in the text until 3 pages later so either should be moved there or the text should be moved to the figure. What does color indicate? Concentration? Explain the figure a little more.

Figure 2 caption: Why is the word “different” used to modify “non-parametric Fisher…”? Different from what? I think “different” can and should be removed.

Line 174: Spell out RMSECV and RMSEC (and note that Table 2 lists RMSEV which I presume should be RMSECV).

Line 195-197: Please describe what is done after the independent PARAFAC models are built from split halves of the data. Are they then compared to each other? To a model on the full set of data? This seems to be the case in your supplemental figures, but don’t make the reader guess.

Line 223: I suppose that it is reasonable to conjecture, or hypothesize, that the four components of the model might correspond to specific chromophores, and it appears to turn out to be true, but it is not at all reasonable to assume that they do. I would just change “assume” to “hypothesize”.

Lines 257-260: No test of normality indicates normality; such tests can only indicate whether a set of data is not consistent with normality. Similarly, there is no way to conclude that a given set of data must “have a non-parametric distribution.” Recommend changing the wording to something like the following: “…and oxidised by-products in EVOOs were consistent with a normal distribution…tocopherols and phenolic compounds respectively were found to be inconsistent with normal distributions. Analysis of variance….”

Line 294: I think by “high classification rates” perhaps you mean “high correct classification rates” or similar. “High classification rates” seems to indicate only that the thing “does a lot of classification” which is kind of meaningless.

Line 298: Where is Table 3? Do you mean Table 2?

Line 299: As noted earlier, I don’t think that the columns in Table 2 describe model parameters but rather measures of model fit.

Line 322: What is PLS-DA? That acronym was not used earlier to my recollection.

Line 326: Replace “discriminate” with “discriminating.”

Supplemental material, Figure S2: Please explain why there are three loadings per component (2 split-half and one…model using all the data?). Is the mismatch of one of the split-half results for the 4th component a problem?

Supplemental material, Figure S3: The caption does not explain what is plotted adequately. What’s the point of this plot? What does the plot tell us?

Author Response

Reviewer 1

Comments and Suggestions for Authors

Comments on “The first identification of the uniqueness and authentication of Maltese Extra virgin olive oil using 3D-fluorescence spectroscopy coupled with multi-way data analysis”

The authors open the Introduction with limitations of certain spectroscopic approaches. The paper needs more context, motivation to open the introduction. Why is this study needed, what is the state of the art prior to this study?

Done as requested we included the current status of the EVOO in the Maltese islands, the need for such study and other research which was done on for determining the authenticity of Maltese EVOOs

There are no recent literature references in this paper, the latest being 2008. Has anyone used these methods more recently? Can you point to similar, more recent studies? Similar either in terms of the statistical methods used (in food provenance/authenticity) or in studies of food oils.

Done as requested we included several recent references and included information regarding the use of EEM-PARAPAC on other food products.  

Page 2, Line 59, “…thar are less robust…” change to “...that are less robust…”

Done

Figure 1. I cannot read the wavelength scales. Use larger font. Render in higher resolution pdf. The image quality in the pdf manuscript I downloaded is terrible, not fit for publication. Maybe my download corrupted the graphical quality, but I don't think so.

Done we fixed the numbering and labelling it should be fine now

Very little discussion of Figure 2. Is it really needed for the paper? Besides it seems to show that the two sample types are not well discriminated (only a handful of data values in each set are clearly outside the bounds of the other set).

We combined PARAFAC to LDA in order word LDA-PARAFAC however the model was only able to discriminate 75% of the data which is a bit on the low side however, considering when the data was used in its pure form without any mathematical transformations and filtering for us is still a significant result. Compared to NPLS we extract far more “component” thus there is more variation in the data that can lead to a higher performing model. In PARAFAC the whole scope was to extract the components which gave the maximum variation for a solution however these may not be most discriminate, that is why we employ NPLS so the extract “component” which also take in consideration the variation in the response this case the geographical origin.

Line 90: Equation should not have X_ijk at the end next to the error term e_ijk (see PARAFAC wiki page, for example)

Done error in copying from one-word document to another

Line 98: “The values along the diagonal…” Diagonal of what matrix or array? Be specific.

Done the values of intensity where changed to zero when the wavelength of excitation was  equal to emission and when intensities derived from scattering which would void the laws of physics where the excitation wavelength was larger than the emission wavelength.

Line 100-101: Either explain terms “split-half validation analysis” and “core consistency” here or note that they are explained later.

Done moved the details where requested

Line 102: Balance between which two parameters? Also I think you mean something like “measures” rather than “parameters.” Split-half validation, core consistency, etc. are measures of “goodness of a model,” not parameters of a model.

Done reworded accordingly

Line 105: What is meant by “mode” here? Could just note that modes are explained later.

I explain since PARAFAC is an extension of a bilinear PCA (not exactly) like in a PCA we would have scores and for the loadings. In PARAFAC we would have loadings in mode 2 and 3 (excitation and emission) and the scores (concentration) and loadings in mode 1 (sample mode).

Line 131-2: Where did the variable Y come from? Do you mean X? And are you saying that X is an array of 0s and 1s? This description of response matrix and classes is confusing. I doubt the comment about the special case of only two classes is needed; clearly that doesn’t come up in your analysis.

As explained just before the original 131-132.: Y is the response matrix i.e the geographical origin of EVOOs whilst X is the full EEM. In our case the Y is composed of 0 and 1 0 being Maltese 1 being non-Maltese. I have reworded the paragraph so that it would become more straightforward and easier to understand.

Line 142: What is meant by “sample mode”? That terms does not seem to have been defined.

Done it corresponds to mode 1 as in PARAFAC

Table 1, column 4: Residual variation increasing with increasing number of components seems odd to me; perhaps I misunderstand what you mean by “residual variation.” I would expect the variance in residuals to decrease as the model fit improves.

Corrected

Line 154-155: It may be worth noting that how these four components were identified as specific fluorescent compounds is described later in the manuscript.

Corrected

Figure 1: This figure is not called out in the text until 3 pages later so either should be moved there or the text should be moved to the figure. What does color indicate? Concentration? Explain the figure a little more.

Corrected

Figure 2 caption: Why is the word “different” used to modify “non-parametric Fisher…”? Different from what? I think “different” can and should be removed.

Corrected

Line 174: Spell out RMSECV and RMSEC (and note that Table 2 lists RMSEV which I presume should be RMSECV).

Corrected

Line 195-197: Please describe what is done after the independent PARAFAC models are built from split halves of the data. Are they then compared to each other? To a model on the full set of data? This seems to be the case in your supplemental figures, but don’t make the reader guess.

Good question. With regards to the PARAFAC model we extract the number of components which best define the solution with regards to the split half validation we compare the two models together in order to seem that the two are similar and from that we extract the % of spilt half similarity such that we can obtain the graphs in the supplementary material. Furthermore, we run the model 10X in order to make sure that we have a stable model. I have included the description in the supplementary Material.

Line 223: I suppose that it is reasonable to conjecture, or hypothesize, that the four components of the model might correspond to specific chromophores, and it appears to turn out to be true, but it is not at all reasonable to assume that they do. I would just change “assume” to “hypothesize”.

Corrected

Lines 257-260: No test of normality indicates normality; such tests can only indicate whether a set of data is not consistent with normality. Similarly, there is no way to conclude that a given set of data must “have a non-parametric distribution.” Recommend changing the wording to something like the following: “…and oxidised by-products in EVOOs were consistent with a normal distribution…tocopherols and phenolic compounds respectively were found to be inconsistent with normal distributions. Analysis of variance….”

You are perfectly right however we have been intrinsically driven to always present normality tests (some of the reviewers might reject the paper if they found that normality test has not been carried out) yet again we had more than 60 samples so technically normality tests could have been avoided.

Line 294: I think by “high classification rates” perhaps you mean “high correct classification rates” or similar. “High classification rates” seems to indicate only that the thing “does a lot of classification” which is kind of meaningless.

Corrected

Line 298: Where is Table 3? Do you mean Table 2?

Corrected

Line 299: As noted earlier, I don’t think that the columns in Table 2 describe model parametersbut rather measures of model fit.

Corrected

Line 322: What is PLS-DA? That acronym was not used earlier to my recollection.

Corrected

Line 326: Replace “discriminate” with “discriminating.”

Corrected

Supplemental material, Figure S2: Please explain why there are three loadings per component (2 split-half and one…model using all the data?). Is the mismatch of one of the split-half results for the 4th component a problem?

Corrected. Minor variations are acceptable I have included the description in the supplementary material

Supplemental material, Figure S3: The caption does not explain what is plotted adequately. What’s the point of this plot? What does the plot tell us?

Corrected included a short description of the plot.

Reviewer 2 Report

The authors report on the development of an analytical method for discrimination of olive oil samples originating from Malta from the samples of other geographic origins, based on the fluorescence spectra. The authors use the Parallel Factor Analysis (PARAFAC) to decompose the 3-way EEMs data into scores, and loadings corresponding to emission and excitation spectra, by finding the optimal number of PARAFAC components which should ideally correspond to the prominent fluorophores in the oil samples. The authors identified that four components are the optimal ones, and the most likely correspond to the content of chlorophyll, oxidation products, tocopherols, and phenolics in oil samples. It seems that the entire PARAAFC analysis was done properly, considering all the performance parameters and validation strategy such as split-half analysis etc.

Furthermore, the scores of the 1st PARAFAC mode (which corresponds to the relative concentration of components/fluorophores in the samples) were further used as input for Analysis of Variance (parametric ANOVA and non-parametric, i.e. Kruskal-Valis), and Fishers LDA, in order to identify which of these components best discriminate among oil samples and to which extent. The authors found that tocopherols and phenolics are the most responsible for discrimination among samples. Also, the LDA confirmed that the correct classification rate is about 70 – 80%. 

Also, the authors used Discriminant Multi-way Partial Least Squares Regression (DM-PLS) on original 3-way EEMs data in order to better identify parts of the fluorescence spectra which can discriminate among the samples of different origin. Based on the VIP scores, the authors have identified the areas of 260 – 380 nm in the excitation mode, and 330 – 480 nm, and 630 – 680 nm in the emission mode, contributing to the discrimination of the samples better than the average, compared to the rest of the spectral range.

The article has been nicely written, and the study was performed correctly. Never the less, I have several major concerns.

Line 95-97. Setting the values to zero, although it is required by the laws of physics, has severe implications on further PARAFAC modelling of EEMs. I hope that the authors took a great precaution with this particular pretreatment. It is much better to set these values to unassigned strings. Please consider re-evaluating your statement!

Figure 1. It is not clear what parts of the figure correspond to the a, b, c, and d. I suppose the order is clockwise.

In addition to the EEMs of PARAFAC components, it would be interesting to see EEMs of some of the representative oil samples. There are 65 of them, but several of them can be provided as a part of Supplementary material. This is the best way to illustrate the extent to which the optimal PARAFAC model fits the experimental data.

Figure 3. Please use the lines of different shapes in addition to different colours and add a legend to the figure.

The authors did not provide details on how the predictive abilities of the DN-PLSR models, as well as LDA models, were assessed. It seems that only cross-validation was performed. The cross-validation parameters such as RMSEcv cannot be used for model prediction assessments because they are used to assess the optimal model complexity (minimum error criteria).

Please indicate that the columns related to training and validation represent the corresponding classification rates in Table 2!

The authors did not provide which cross-validation resampling strategy was used (leave-one-out, Venetian blinds, continuous blocks etc., and which data split ratio was used in the case of the last two).

In addition to calibration, and cross-validation samples, please form a separate, independent test/prediction set (approximately 25% of the total number of samples, using Kennard Stone algorithm or another way for selecting a representative sample subset) and run the LDA and DM-PLS to accurately estimate predictive ability of the models.

Author Response

Reviewer 2

The authors report on the development of an analytical method for discrimination of olive oil samples originating from Malta from the samples of other geographic origins, based on the fluorescence spectra. The authors use the Parallel Factor Analysis (PARAFAC) to decompose the 3-way EEMs data into scores, and loadings corresponding to emission and excitation spectra, by finding the optimal number of PARAFAC components which should ideally correspond to the prominent fluorophores in the oil samples. The authors identified that four components are the optimal ones, and the most likely correspond to the content of chlorophyll, oxidation products, tocopherols, and phenolics in oil samples. It seems that the entire PARAAFC analysis was done properly, considering all the performance parameters and validation strategy such as split-half analysis etc.

Furthermore, the scores of the 1st PARAFAC mode (which corresponds to the relative concentration of components/fluorophores in the samples) were further used as input for Analysis of Variance (parametric ANOVA and non-parametric, i.e. Kruskal-Valis), and Fishers LDA, in order to identify which of these components best discriminate among oil samples and to which extent. The authors found that tocopherols and phenolics are the most responsible for discrimination among samples. Also, the LDA confirmed that the correct classification rate is about 70 – 80%. 

Also, the authors used Discriminant Multi-way Partial Least Squares Regression (DM-PLS) on original 3-way EEMs data in order to better identify parts of the fluorescence spectra which can discriminate among the samples of different origin. Based on the VIP scores, the authors have identified the areas of 260 – 380 nm in the excitation mode, and 330 – 480 nm, and 630 – 680 nm in the emission mode, contributing to the discrimination of the samples better than the average, compared to the rest of the spectral range.

The article has been nicely written, and the study was performed correctly. Never the less, I have several major concerns.

Line 95-97. Setting the values to zero, although it is required by the laws of physics, has severe implications on further PARAFAC modelling of EEMs. I hope that the authors took a great precaution with this particular pretreatment. It is much better to set these values to unassigned strings. Please consider re-evaluating your statement!.

Thanks a lot for this comment we actually tested both seniors however substituting missing EEM data with ‘NaN’ led to inconsistent models which converged only after a large number of iterations. On the other hand, more consistent and quickly converging models were obtained when the missing data with the exception of the Raman scattering range was filled with zeros similar to what was done by Christensen et al., 2005.

Figure 1. It is not clear what parts of the figure correspond to the a, b, c, and d. I suppose the order is clockwise.

Corrected

In addition to the EEMs of PARAFAC components, it would be interesting to see EEMs of some of the representative oil samples. There are 65 of them, but several of them can be provided as a part of Supplementary material. This is the best way to illustrate the extent to which the optimal PARAFAC model fits the experimental data.

We included a typical EEM of a typical EVOO for comparative reason. 

Figure 3. Please use the lines of different shapes in addition to different colours and add a legend to the figure.

Corrected

The authors did not provide details on how the predictive abilities of the DN-PLSR models, as well as LDA models, were assessed. It seems that only cross-validation was performed. The cross-validation parameters such as RMSEcv cannot be used for model prediction assessments because they are used to assess the optimal model complexity (minimum error criteria).

Corrected we employed the use of an additional independent DA in order to determine the predictability of the model as suggested by your last comment.

.

Please indicate that the columns related to training and validation represent the corresponding classification rates in Table 2!

 Corrected

The authors did not provide which cross-validation resampling strategy was used (leave-one-out, Venetian blinds, continuous blocks etc., and which data split ratio was used in the case of the last two).

 Corrected

In addition to calibration, and cross-validation samples, please form a separate, independent test/prediction set (approximately 25% of the total number of samples, using Kennard Stone algorithm or another way for selecting a representative sample subset) and run the LDA and DM-PLS to accurately estimate predictive ability of the models.

Corrected results presented thanks for bringing it to our attention. Details are as follow with regards to the independent LDA model a Venetian blind cross validation was carried out such that the whole dataset into two sets the training and test sets (the former to build the model, the latter to validate its predictability). The following method was adopted in order to cover as such variation in the two sets and at the same time being able to compare the outcomes after the different pretreatments. The Maltese and the non-Maltese samples were grouped in an ascending way so that the first 30 samples would represent Maltese EVOO’s whilst the rest correspond to non –Maltese EVOO’s. A Venetian blinds cross-validation which selects every sth sample from the data by making data splits such that all samples are left out exactly once (s=5).  This sampling method excluded 20% of the observation so that they would be retained as the testing set. The remaining 80% of the observation were used as to build the training set

Reviewer 3 Report

The topic is related to the aims and scope of “Foods”.

Authors presented the application of fluorescence spectroscopy coupled with advanced data analysis for the identification of olive oil origin. The data analysis is appropriately designed but problematic is that the work relates to the state of knowledge of the mid-2000s. The latest cited work comes from 2009. A quick check in the Scopus shows that there are 286 records for keywords "olive oil origin", of which 220 are newer than 2009. Contemporary solutions (chemistry+data processing) allow to discriminate exact geographical origin, cultivars, effects of harvesting year etc. with clearly separated groups (mainly using UV-VIS-NIR-MIR, chromatography). Fluorescence plays important role rather for assessment of qualitative and quantitative properties of olive oil samples and screening for adulteration with low-grade oils. Thus Fig.2 shows, that the conclusion about reliable method is exaggerated, as points of Maltese origin and non-Maltese origin are mixed and is difficult to draw the discrimination lines between sets of data.

However the subject is very interesting for economy and scientific community, thus I would like to encourage Authors to update the references to contemporary and re-design the experiment with any coupled technique. 

Author Response

Reviewer 3 (MZM)

We are aware of the contemporary solutions available and we have extended this method to identify local and foreign varieties in this field and this goes beyond the extent of the current use identified by the 3rd reviewer. 

We note the suggestion re the current use of techiques (UV-VIR-NIR-MIR, chromatography)  to discriminate geographic origin, cultivars and effect of harvesting year. However the relative time-consuming nature of these methods, the expense in terms of bespoke equipment, columns, solvents, labour and the restricted use in the laboratory to separate and quantify phenolic, tocopherols, chlorophylls and oxidised compounds should be considered. In contrast, we wish to highlight that PARAFAC analysis takes less than 20 minutes, is still sensitive and enables the development of field-guided equipment (already on the market) to determine both the quality and quantity of the product.

Moreover, while it is acknowledged that traditional methods (involving peak to peak identification followed by quantification) are easy when a small number of compounds with standards are available, the sheer complexity of the situation when a complex matrix with over 40 chemicals of different compounds is involved should not be overlooked.

Therefore, the information presented in this paper, constitutes a significant advantage to the current state-of-the-knowledge in the field. For these reasons, we believe that these findings should be disseminated to the scientific community.

Furthermore, it is imperative to note that we inserted Figure 2 (LDA analysis bi-plot) to validate our claim that LDA was NOT as effective as DN-PLSR and not to give the conclusion that LDA is effective at separating EVOOs of Maltese and non-Maltese origin, as erroneously claimed by the 3rd reviewer. Indeed our paper stressed the use of DN-PLSR (correct classification close to 100%) rather than LDA (see discussion and conclusions). Hence we reiterate the validity of the science presented.

Round 2

Reviewer 1 Report

The authors' response to my review is acceptable. I cannot judge well Reviewer 3's critique of the paper nor the authors' response.

Reviewer 2 Report

Dear authors,

I congratulate you on all efforts to make this paper a high quality work. 

I completely agree with all the corrections introduced and responses provided. Therofe, I recommend the article to be published in the present form. 

Reviewer 3 Report

Required corrections have been made.